# Randomised feasibility trial of the helping families programme-modified: an intensive parenting intervention for parents affected by severe personality difficulties

Crispin Day  ,[1,2] Jackie Briskman,[1] Mike J Crawford,[3] Lisa Foote,[4] Lucy Harris,[2] Janet Boadu,[5] Paul McCrone,[5,6] Mary McMurran,[7] Daniel Michelson,[8] Paul Moran,[9] Liberty Mosse,[1] Stephen Scott,[10] Daniel Stahl,[11] Paul Ramchandani,[12] Tim Weaver[13]

For numbered affiliations see end of article.

**Correspondence to**
Dr Crispin Day;
crispin.1.day@kcl.ac.uk

## ABSTRACT

**Background** Specialist parenting intervention could improve coexistent parenting and child mental health difficulties of parents affected by severe personality difficulties.

**Objective** Conduct a feasibility trial of Helping Families Programme-Modified (HFP-M), a specialist parenting intervention.

**Design** Pragmatic, mixed-methods trial, 1:1 random allocation, assessing feasibility, intervention acceptability and outcome estimates.

**Settings** Two National Health Service health trusts and local authority children's social care.

**Participants** Parents: (i) primary caregiver, (ii) 18 to 65 years, (iii) severe personality difficulties, (iv) proficient English and (v) capacity for consent. Child: (i) 3 to 11 years, (ii) living with index parent and (iii) significant emotional/behavioural difficulties.

**Intervention** HFP-M: 16-session home-based intervention using parenting and therapeutic engagement strategies. Usual care: standard care augmented by single psychoeducational parenting session.

**Outcomes** Primary feasibility outcome: participant retention rate. Secondary outcomes: (i) rates of recruitment, eligibility and data completion, and (ii) rates of intervention acceptance, completion and alliance (Working Alliance Inventory-Short Revised). Primary clinical outcome: child behaviour (Eyberg Child Behaviour Inventory). Secondary outcomes: child mental health (Concerns About My Child, Child Behaviour Checklist-Internalising Scale), parenting (Arnold-O'Leary Parenting Scale, Kansas Parental Satisfaction Scale) and parent mental health (Symptom-Checklist-27). Quantitative data were collected blind to allocation.

**Results** Findings broadly supported non-diagnostic selection criterion. Of 48 participants recruited, 32 completed post-intervention measures at mean 42 weeks later. Participant retention exceeded a priori rate (HFP-M=18; Usual care=14; 66.7%, 95% CI 51.6% to 79.6%). HFP-M was acceptable, with delivery longer than planned. Usual care had lower alliance rating. Child and parenting outcome effects detected across trial arms with potential HFP-M advantage (effect size range: 0.0 to 1.3).

**Conclusion** HFP-M is an acceptable and potentially effective specialist parenting intervention. A definitive trial is feasible, subject to consideration of recruitment and retention methods, intervention efficiency and comparator condition. Caution is required in interpretation of results due to reduced sample size. No serious adverse events reported.

**Trial registration number** ISRCTN14573230

## Strengths and limitations of this study

► This randomised trial assessed the feasibility of a specialist parenting intervention for coexistent mental health problems of parents affected by severe personality difficulties and their children.

► Findings provide useful evidence to support further evaluation of this specialist parenting intervention within a definitive trial, with modifications required to improve intervention efficiency, augmented usual care condition acceptability and participant enrolment and retention.

► Caution is required in interpretation of results due to reduced sample size.

► The trial population's complex personality difficulties underline the importance of effective and sensitive management of trial consent procedures, random allocation and ongoing engagement of participants, particularly for those allocated to the usual care condition.

## INTRODUCTION

Mental ill health is the largest cause of disability, with three-quarters of lifetime disorders starting during childhood.[1] Children of parents affected by severe personality difficulties are at particular risk due to the impact of parents' symptoms and associated

impairment on parenting capacity.[2–4] Lack of evidence-based treatments, underdeveloped care pathways and stigma result in poorer immediate and longer-term child outcomes, increase intergenerational transmission of mental health difficulties and perpetuate social disadvantage.[5 6]

Severe personality difficulties, including personality disorders, affect over 4% of UK adults and 40% of mental health service users, at least one-quarter of whom are parents.[7 8] Per annum UK treatment costs exceed £70 million, with wider societal costs estimated at £8 billion per year.[9] Characterised by problematic interpersonal relationships, emotional dysregulation and poor impulse control, severe personality difficulties are associated with insensitive and intrusive interactions with offspring, family hostility, inconsistent and unpredictable family routines that undermine affectionate, stable and responsive parenting required for healthy child development.[2–4] Affected parents are likely to suffer higher parenting stress and lower satisfaction, exacerbating underlying mental health difficulties.[10 11]

One in 10 UK children suffer mental health disorders, with children of parents with severe personality difficulties at substantially higher risk of intergenerational transmission, most commonly behavioural disorders.[12 13] These childhood disorders are associated with academic failure, school exclusion, maltreatment, self-harm and gang affiliation, with increased life-time risk of comorbid mental and physical health conditions, drug misuse, offending and worklessness.[1 12 13] Annual UK public service costs for severe behavioural problems are estimated at £5000 per child including £1400 health costs. Lifetime estimated costs range from £85 000 (moderate case) to £260 000 (severe case).[14 15]

Concerted preventative and early intervention during pregnancy, infancy and childhood is warranted.[16] Effective care models for co-occurring child and parental mental health problems are significant health policy and research priorities.[17–19] Nevertheless, routine care is highly variable, generally focussed on the needs of *either* the adult *or* the child,[17] resulting in under-identification, poor understanding of the interrelationship between child and parent difficulties and misattribution of parenting difficulties to adult mental health symptoms per se.[20] Affected parents can be reluctant to engage due to stigma, treatment scepticism and the interpersonal difficulties and adverse life circumstances associated with personality difficulties.[21]

Parenting and parent factors are central to much effective child mental health treatment and problem remediation. Parenting programmes are the recommended cost-effective treatment for child behaviour problems.[22 23] However, non-specialised parenting programmes, not specifically designed for parents affected by mental health difficulties, often result in poorer engagement, acceptability and outcomes.[24 25] Specialist interventions for some parental mental health conditions, such as depression, substance misuse and eating disorders, have demonstrated improved outcomes and reduced intergenerational transmission, by up to 40%.[26 27] Specialist programmes for parents affected by personality difficulties are at an earlier stage of evidence production.

Helping Families Programme-Modified (HFP-M) was developed as a specialist, intensive parenting intervention to address this need and service gap. It aims to improve immediate child and parenting outcomes with longer-term potential to reduce intergenerational transmission and psychosocial adversity within affected families.[4] Guided by recommended frameworks, HFP-M development synthesised two existing evidence-based interventions, incorporating relevant clinical practice recommendations and service user consultation.[22 28–33] Consistent with other promising programmes aiming to improve parenting and child outcomes in high risk groups, HFP-M is based on a transtheoretical model of parenting drawing on attachment, social learning and cognitive-affective theories and methods.[34 35] HFP-M does not target personality difficulties per se but aims to improve the ways that these characteristics affect parenting behaviour, emotional regulation, parent-child relationships and lead to adverse child outcomes.

HFP-M has three structured components:[36] (i) Core Therapeutic Process: including partnership and goal-based methods to promote collaborative relational engagement, shared formulation, empathic parent validation and crisis management;[36] (ii) Parent Groundwork: including emotion-focussed, cognitive, behavioural and interpersonal strategies to manage parental emotional dysregulation and hostility while relating to their children and undertaking parenting tasks and (iii) Parenting Strategies: including consistent use of positive parenting skills, such as, praise, consequences and limit setting, and relational and affective parenting methods such as emotionally responsive, warm care-giving and reflective function.

Definitive trials require successful recruitment, retention and intervention acceptability.[30 37–41] Participant retention underpins trial validity. Lower rates reduce power, undermine interpretation of findings and increase costs. HFP-M retention, and initial recruitment was expected to be affected by: (i) participants' core clinical features, (ii) greater exposure to family stress, negative life events, lower levels of social support and comorbid mental health conditions and (iii) under-identification of need within routine care, negative referrer expectancies, lower service engagement and attendance. These factors are reflected in evidence derived from 45 personality disorder treatments trials reported in two systematic reviews indicating a median participant non-completion rate of 35%.[37 42] Pre-feasibility trial case series findings, consultation with service user, clinicians and research ethics indicated that initial plans for trial recruitment based on personality disorder research diagnosis was unlikely to be viable for practical, participant acceptability and ethical reasons.[21]

To be useful and effective in practice, interventions need to demonstrate both clinical efficacy and user acceptability. Acceptability refers to service user judgements across four inter-related domains of intervention satisfaction: (i) relevance, (ii) content and procedures, (iii) clinician/provider characteristics and (iv) outcome suitability.[39 40]

This study reports quantitative findings from a randomised feasibility trial of HFP-M, based on a published protocol.[4] The trial aimed to assess research and clinical feasibility of HFP-M for a target population with coexisting parent personality difficulties and child mental health difficulties with findings being used to inform the design of a full-scale trial.

The primary feasibility outcome was a participant retention rate of at least 65% post-intervention. Secondary feasibility outcomes were rates of: (i) participant identification and recruitment, (ii) data collection and (iii) intervention use, uptake and acceptability. Primary clinical outcome was child behaviour. Secondary clinical outcomes included parental child concerns, child internalising difficulties, parenting behaviour, satisfaction and psychological well-being. The trial sought to produce effect sizes and variance estimates for child and parent outcomes necessary to power a full-scale trial.

The findings of a parallel qualitative process evaluation investigating the influence of contextual factors on trial and intervention implementation and outcome generation are published elsewhere, as are the full findings of preliminary intervention costs and estimates of cost-effectiveness.[43]

## METHOD
### Design
Mixed-method, two-arm, parallel feasibility trial with random allocation in a 1:1 ratio to either: (i) HFP-M, or (ii) Usual care.

Quantitative data were collected at pre-randomisation baseline (Time 1); post-intervention (Time 2), 6 months from baseline and follow-up (Time 3), 4 months and 10 months from baseline.

### Eligibility criteria
Parent: (i) primary parental caregiver for index child, (ii) aged 18 to 65, (iii) experiencing severe personality difficulties, assessed by self-administered Standardised Assessment of Personality-Abbreviated Scale (SAPAS) score of ≥3, the optimal cut-point for the intended sample population,[44] (iv) proficient written and spoken English and (v) capacity to provide informed consent.

Child: (i) aged 3 to 11, (ii) living with index parent and (iii) experiencing significant emotional/behavioural difficulties, assessed by Strengths and Difficulties Questionnaire Total Score of ≥17.[45]

#### Exclusion criteria
Parent: (i) coexisting psychosis, (ii) engagement in another structured parenting intervention, (iii) inpatient status or (iv) insufficient language/cognitive abilities.

Child: (i) pervasive developmental disorder, (ii) not residing with index parent or (iii) considered for/subject to child protection supervision.

## Interventions
### HFP-M intervention
Sixteen-session home-based 1:1 parenting intervention for parents with severe personality difficulties, including personality disorder. Session modules proceeded iteratively to (i) establish effective, validating collaborative partnership, (ii) develop shared understanding of severe parental personality difficulties' impact on parenting, child functioning and family ecology, (iv) implement parent quick wins, parenting and child intervention goals, (v) use evidence-based parenting and parent self-care strategies to achieve agreed goals and (vi) recurrent review of goals and therapeutic partnership. Six trial therapists received eight, 3-hour, training sessions provided by HFP-M programme developers and clinical experts. Trial therapists completed structured checklists and received fortnightly supervision from experienced HFP-M clinicians to support clinical implementation and fidelity.

### Usual care
No systematised parenting pathway was typically provided for the participant population. To provide consistent, low intensity support, participants could receive an additional home-based one-to-one parent information and support session. Derived from the evidence-based Empowering Parents Empowering Communities parenting programme,[46] session content included: (i) brief exploration of parenting and child needs, family support, parent priorities and goals, and (ii) focus on one parent priority topic, selected from Being Good Enough, Listening to My Child, Praising My Child, Taking Care of Myself, Understanding My Child's Behaviour, My Child's Emotion or Playing Together. The additional session was delivered by three trained parent practitioners, who received ongoing supervision to support implementation and fidelity.

### Concomitant interventions
Both HFP-M and the single Usual care parent support session were provided in addition to existing medical, psychosocial and educational support and treatment services used by participating parents and their families. A joint-working protocol specified procedures for care coordination and information sharing between trial therapists and routine services.

## Measures
### Participant characteristics
Descriptive data were collected on parent and child age, gender and ethnicity, family household composition, participant diagnostic status and family socioeconomic status. Data from EuroQol five dimension scale (EQ-5D),[47 48] provided information about parent and child health and disability.

## Feasibility evaluation

Structured record sheets, completed prospectively by research staff and trial therapists, documented: (i) participant identification and verbal consent, (ii) screening, eligibility, informed written consent, randomisation and reasons for non-participation, and (iii) data collection and missing data.

## Clinical outcomes

► **Eyberg Child Behaviour Inventory (ECBI)**,[49] a 36-item questionnaire assessing intensity and number of disruptive behaviour problems in 2 to 16 year-olds, providing a comprehensive measure of child behaviour difficulties. Intensity Scale score of ≥131 indicates significant severity. Problem Scale score of ≥15 indicates significant number of problems.

► **Concerns About My Child (CAMC)**,[46] a Visual Analogue Scale (0 to 100) rating three parental concerns about their child. Concerns nominated at baseline were re-rated at each time point, providing a sensitive, individualised index of change.

► **Child Behaviour Checklist-Internalising Scale (CBCL-Int)**,[50] a 32-item questionnaire assessing internalising problems in 6 to 18 year-olds, with an alternate 36-item version for children aged 1½ to 5 years. Standardised T-scores combine results from both versions. A score of ≥60 indicates clinical caseness.

► **Arnold-O'Leary Parenting Scale (PS)**,[51] a 30-item questionnaire assessing dysfunctional parental discipline behaviour for children aged 2 to 16 years, which correlates with more time-consuming observational ratings. A score of ≥3.2 differentiates clinic and non-referred children.

► **Kansas Parental Satisfaction Scale (KPSS)**,[52] a 3-item scale providing a brief measure of parenting stress and satisfaction.

► **Symptom-Checklist-27**,[53] a 27-item questionnaire assessing psychological symptoms in adults that provides a Global Severity Index of psychopathology.

## Intervention acceptability

► **Working Alliance Inventory-Short Revised (WAI-SR)**,[54] a parent completed 12-item questionnaire assessing therapeutic relationship quality consisting of three subscales: (i) Goals, measuring agreement on intervention goals and outcomes, (ii) Tasks, measuring agreement on behaviours and thoughts underpinning intervention process and (iii) Bond, measuring mutual trust, acceptance and confidence.

► Structured worksheets recorded intervention uptake, attendance, retention, reasons for missed sessions and dropout.

## Health economic

► **Client Service Receipt Inventory (CSRI)**,[55] a schedule adapted to measure the use of services by caregivers and children.

► **EQ-5D-5L and EQ-5D-Y**,[47 48] a generic measure of health-related quality of life used to generate quality-adjusted life years. EQ-5D-Y is adapted for younger respondents.

## Sample size

The primary feasibility criterion was post-intervention retention rate of at least 65%.[37 42] A CI approach was used to calculate a planned sample size of n=70.[56] Using a 95% CI for the proportion of parents who completed treatment and an expected completion rate of 80% based on previous evaluations of HFP, it was determined that an HFP-M intervention sample size of n=35 would provide a sufficiently precise estimate (95% CI 0.67 to 0.93). A sample size of n=70 was also sufficient to obtain stable estimates of population variances for future power calculations.[57]

## Recruitment and consent procedures
### Settings

Recruitment took place in two large UK National Health Service (NHS) health services and concomitant local authority children's social care service in London (Site 1 and 2), located in areas of high mental health morbidity.

### Identification and consent

Clinical keyworkers undertook exploratory discussion and provided written information to potential participants. With consent, contact details were provided to researchers, who provided information about study aims, eligibility criteria, procedures and a Participant Information Sheet. One week later, parents were contacted to determine participation and obtain written informed consent.

### Allocation and randomisation

Participants were allocated a unique, anonymised ID number and randomised to trial conditions between 11 May 2016 and 29 March 2017 by Clinical Trials Unit, King's College, London. Allocation was communicated confidentially to the trial coordinator, other researchers remained blind to allocation.

## Data collection
### Screening and assessments

Following consent, parents completed screening measures, and, when eligible, baseline measures in a standard sequence. Any parent discomfort was addressed sensitively and supportively. Persistent, non-intrusive efforts were used to complete post-intervention and follow-up data collection. Participants were reimbursed £10 per hour for data completion at each point.

## Analysis

Statistical analysis was mainly descriptive using means and SD for continuous data, or medians and range where data were skewed. Frequencies and proportions were used to describe categorical variables. Feasibility of trial retention was assessed using the proportion of a predetermined

parameter and estimated 95% CIs. Clinical outcomes were analysed using analysis of covariance models to estimate likely range of intervention effect, by assessing 95% CI, at post-treatment, with pre-randomisation values as a covariate.[58] Follow-up data were not included in these analyses due to a smaller sample than planned. Standardised effect sizes were calculated using Cohen's d. Given the complexity of coexistent parent and child mental health difficulties, the smallest change in outcome identified as clinically important is equivalent to a small effect size. Population variances for future power calculations were determined using the upper 80th percentile of CIs around the estimated population variance.[58]

## Patient and public involvement

A senior staff member of a national service user organisation was a co-applicant and contributed to research conception, planning and governance. A service user researcher was involved in the analysis, interpretation and dissemination of findings. A service user panel advised on trial planning, intervention methods, outcome selection and interpretation of findings.[59]

## Independent NHS research and development approval

Research and development approvals were obtained from South London and Maudsley NHS Foundation Trust and Central and North-West London NHS Foundation Trust.

## Serious adverse events

The chief investigator was responsible for reporting serious adverse events to the trial's independent Data Monitoring and Ethics Committee and responsible research ethics committee. No serious adverse events were reported.

## RESULTS
### Feasibility evaluation

Recruitment took 4 months longer than planned due to delays in Site 2, resulting in a revised sample size of 48. Obtaining keyworker information on service users approached about trial participation proved impractical.

All referred service users (n=89, 100.0%) consented to research contact (see figure 1). Adult mental health services (AMHS) referred 30 (33.7%) parents, child and adolescent mental health services (CAMHS) 29 (32.6%) and children's social care (CSS) 30 (33.7%). Site 1 referred 65 (73.0%) parents. Researchers made contact with 87 (97.7%) parents, requiring 1 to 13 communications per participant.

Sixty (69.7%) parents met initial criteria and completed screening. The most common reason for ineligibility was parents declining trial participation (n=12, 13.5%). Six parents were excluded due to child-related reasons, most commonly presence of child developmental disorder.

Forty-eight consenting participants met parent and child screening criteria, all of whom completed baseline measures, representing 80.0% of screened parents, 53.9%

of referred parents and 68.8% of the planned sample. Five (8.3%) met neither parent nor child screening criteria, four (6.7%) did not meet SAPAS criterion and three (5.0%) did not meet Strengths and Difficulties Questionnaire (SDQ) criterion.

Thirty-six (75.0%) participants were from Site 1, exceeding the site recruitment target. Site 2 recruitment (n=12) was 34.3% of that planned, due to delayed recruitment and lower service engagement. Eighteen (37.5%) participants were referred by AMHS, 16 (33.3%) CAMHS and 14 (29.2%) CSS.

There was a significant difference in trial condition uptake (HFP-M: n=21, 87.5%; Usual care n=15, 62.5%; $\chi^2$=4.0, df=1 (48), p<0.05). Modal duration between randomisation and starting HFP-M was 2 weeks (range 1 to 23 weeks). Parents declined HFP-M because one gave birth, another was in couple conflict about participation and a third did not respond to persistent contact (see figure 1). The modal duration before Usual care parenting session receipt was 5.5 weeks (range 1 to 17 weeks). The most common reason for declining the additional Usual care parenting session was that participants hoped to be allocated to HFP-M.

Thirty-two (66.7%, 95% CI 51.6% to 79.6%) participants completed post-intervention measures. The majority of participants who did not take up the intervention offered, mainly in the Usual care condition, did not complete post-intervention measures.

Twenty-one (65.6%) parents completing post-intervention measures were also assessed at follow-up, representing 43.8% of all participants. Post-intervention measures were completed a mean of 42.0 weeks (SD 14.6) after baseline, due to HFP-M delivery being lengthier in duration than planned. Follow-up measures were completed a mean of 20 weeks (SD 8.9) later. Researchers required one to eight participant contacts to arrange post-intervention and follow-up data collection. Main reasons for non-completion included unable to contact participant, participant declining completion and participant health and life circumstances.

There were less than 0.1% missing data items across clinical measures at each time point.

### Sample characteristics
#### Demographic characteristics

All 48 participants were the biological parent of the index child, one was a father (2.1%) (see table 1). The majority were lone parents (n=31, 67.4%), mean age 34.9 (SD 7.1) years. The majority (n=28, 61.0%) were White British/White, with fewer Black/Black British (n=9, 19.6%) and Dual Heritage (n=6, 13.5%) parents. English was the most common first language (n=44, 91.7%). Twenty-five (55.6%) participants completed education at 18 years or younger. Four participants (8.9%) were higher education graduates.

Most participants (n=37, 80.4%) were not in paid employment. Nine (19.5%) had partners in paid employment, predominantly part-time (n=7, 15.2%). No parent was in paid employment in 30 (65.2%) households. Twenty-five

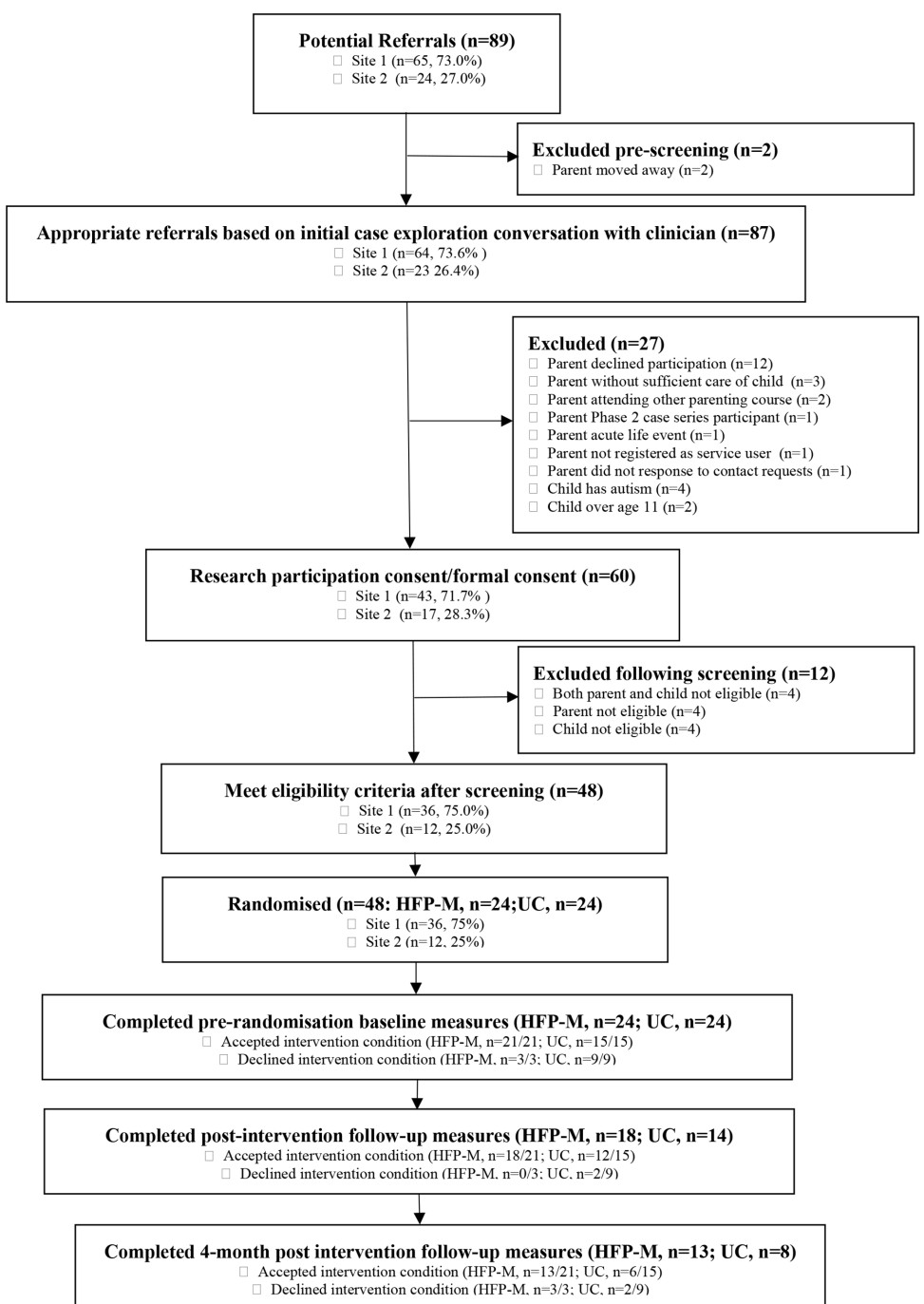

**Figure 1** Consolidated Standards of Reporting Trials diagram randomised feasibility trial recruitment and retention. HFP-M, Helping FamiliesProgramme-Modified; UC, usual care.

(86.2%) parents had received a formal psychiatric diagnosis, mean time since initial diagnosis was 9.7 (7.4) years.

Twenty-six (54.2%) children were male, mean age of 7.8 (SD 2.2) years. Median number of children in the home was 2, range 1 to 5 (see table 1). Participants reported significant difficulties with anxiety/depression, pain/discomfort, with a smaller proportion experiencing difficulties in undertaking everyday activities (see table 1).

## Clinical characteristics
Over 80 per cent of ECBI Problem and Intensity scores at baseline exceeded the clinical caseness cut-off, with

similar caseness rates for CBCL-Int and PS (see table 2). The most common baseline CAMC child-related parent concerns were conduct, self-regulation, parent-child relationships and emotional distress (see table 3).

## Trial interventions acceptability
### HFP-M attendance
Of 21 participants who accepted HFP-M, 13 (61.9%, 95% CI 38.4% to 81.9%) completed HFP-M within the trial period. Six (28.6%, 95% CI 11.3% to 52.2%) withdrew before completion due to acute adult mental health crisis and complex family circumstances unrelated to

**Table 1** Participant demographic characteristics

| Demographic characteristics | Baseline (Time 1) | | |
| --- | --- | --- | --- |
| | Total | Intervention | Usual care |
| Parent gender (female) (n, %) | 47 (97.9) | 24 (100.0) | 23 (95.8) |
| Parent age (yrs.) (mean, SD) | 34.9 (7.1) | 34.7 (7.5) | 35.0 (6.9) |
| Received psychiatric diagnosis (n, %) | 25 (86.2) | 16 (88.9) | 9 (81.8) |
| Psychiatric diagnosis duration (yrs.) (mean, SD) | 9.7 (7.4) | 8.6 (7.0) | 10.9 (8.1) |
| Parent relationship to index child (biological parent) (n, %) | 46 (100.0) | 24 (100.0) | 22 (100.0) |
| Index child gender (male) (n, %) | 26 (55.3) | 12 (50.0) | 14 (60.9) |
| Index child age (mean, SD) | 7.8 (2.2) | 7.7 (2.0) | 7.9 (2.2) |
| Number of children at home (median, range) | 2 (1–5) | 2 (1–5) | 2 (1–5) |
| Not in paid employment (n, %) | 37 (80.4) | 19 (79.2) | 18 (81.8) |
| Lone parent (n, %) | 31 (67.4) | 14 (58.3) | 17 (77.3) |
| Partner in employment (n, %) | 9 (19.5) | 6 (25.0) | 3 (12.5) |
| Parent education (n, %) | | | |
| Graduate | 4 (8.9) | 2 (8.3) | 2 (9.5) |
| University not completed | 3 (6.7) | 3 (12.5) | 0 (0.0) |
| Other for example, NVQ | 13 (28.9) | 6 (25.0) | 7 (33.3) |
| Left school 18 years | 9 (20.0) | 4 (16.7) | 5 (23.8) |
| Left school 16 years | 7 (15.6) | 4 (16.7) | 3 (14.3) |
| Left school under 16 | 9 (20.0) | 5 (20.9) | 4 (19.0) |
| Ethnicity (n, %) | | | |
| White UK/White other | 28 (61.0) | 13 (54.2) | 15 (68.2) |
| Black UK/African Caribbean | 9 (19.6) | 7 (29.2) | 2 (9.1) |
| Dual heritage | 6 (13.0) | 3 (12.5) | 3 (13.6) |
| Black UK/African | 2 (4.3) | 0 (0.0) | 2 (9.1) |
| Other | 1 (2.2) | 1 (4.2) | 0 (0.0) |
| Parent health status* (n, %) | | | |
| Mobility problems | 9 (25.0) | 4 (23.5) | 5 (26.4) |
| Problems in self-care washing and dressing | 3 (8.3) | 1 (5.9) | 2 (10.5) |
| Difficulties in undertaking usual activities | 13 (36.1) | 7 (41.1) | 6 (31.6) |
| Suffered pain/discomfort | 17 (47.2) | 6 (35.3) | 11 (57.9) |

*EQ-5D-5L health moderate/severe status.
NVQ, National Vocational Qualification; yrs., years.

HFP-M receipt. Recruitment delay and longer than anticipated intervention duration resulted in two (9.5%, 95% CI. 1.2% to 30.4%) participants not fully completing HFP-M before trial conclusion. The mean number of HFP-M appointments offered was 15.8 (SD 7.7) and mean number attended was 11.2 (SD 6.3). HFP-M appointment

**Table 2** Participant baseline clinical caseness

| Measure | Baseline | | |
| --- | --- | --- | --- |
| | Total | Intervention | Usual care |
| ECBI problem caseness (≥15) (n, %) | 41 (89.1) | 21 (87.5) | 20 (90.9) |
| ECBI intensity caseness (≥131) (n, %) | 39 (83.0) | 19 (79.2) | 20 (83.3) |
| CBCL-Int (t-score) caseness (≥60) (n, %) | 45 (95.8) | 23 (95.8) | 22 (95.7) |
| PS caseness (≥3.2) (n, %) | 35 (74.5) | 18 (75.0) | 17 (73.9) |

CBCL-Int, Child Behaviour Checklist-Internalising Scale; ECBI, Eyberg Child Behaviour Inventory; PS, Arnold-O'Leary Parenting Scale.

**Table 3** Parent-reported concerns about index child

| Concern category | Primary | Secondary | Tertiary | Total |
|---|---|---|---|---|
| Conduct problems* (n, %) | 18 (38.3) | 27 (58.7) | 24 (53.3) | 69 (50.0) |
| Parent-child relationship and communication (n, %) | 16 (34.0) | 2 (4.4) | 4 (8.9) | 22 (15.9) |
| Child self-regulation† (n, %) | 4 (8.5) | 11 (23.9) | 7 (15.6) | 22 (15.9) |
| Emotional distress‡ (n, %) | 8 (17.0) | 4 (8.8) | 7 (15.6) | 19 (13.8) |
| Other§ (n, %) | 1 (2.1) | 2 (4.4) | 1 (2.2) | 4 (2.9) |
| School (n, %) | 0 (0.0) | 0 (0.0) | 2 (4.4) | 2 (1.5) |
| Total (n, %) | 47 | 46 | 45 | 138 |

*including anger, tantrums, defiance, non-compliance, aggression, running away and lying.
†including overactivity, poor concentration, overeating and wetting.
‡including low mood, anxiety, low self-esteem.
§including risk behaviours.

attendance was 70.2%. Mean duration of HFP-M delivery was 28.4 (SD 21.7) weeks.

### HFP-M therapeutic alliance acceptability

Eighteen (85.3%) HFP-M participants completed post-intervention WAI-SR (mean total score=73.8, SD 10.4). Mean subscale scores were consistently in the upper end of the scale (Mean Tasks Subscale score=24.9, SD 3.5; Bond Subscale score=24.9, SD 3.9; Goals Subscale score=23.9, SD 4.1).

### Usual care

All 15 participants accepting the additional parenting session completed. Due to lower retention, only six participants provided post-intervention WAI-SR data (mean total score=56.2, SD 18.8). Mean subscales scores were consistently lower than the HFP-M condition (Mean Tasks Subscale score=17.4, SD 7.2; Bond Subscale score=19.8, SD 7.1; Goals Subscale score=19.0, SD 7.6).

There appeared to be a substantial difference in WAI-SR scores between the two conditions but there was insufficient Usual care data to test for a statistical difference.

No adverse events were reported during the trial. Participant intervention withdrawal most frequently occurred due to deterioration in participant mental health and life circumstances, unrelated to trial participation.

### Clinical outcomes

There were estimated mean improvements from baseline scores across a number of outcomes within both trial conditions (see table 4). HFP-M mean differences exceeded those in Usual care on several outcomes. These findings should be treated with caution given the wide CIs.

Estimated effect sizes showed a general post-intervention advantage for HFP-M on a range of outcomes. Medium effects for child behavioural problem severity (ECBI Intensity, effect size (ES) 0.4, CI −0.3 to 1.1), and parenting satisfaction (KPSS, ES 0.4, CI −0.3 to 1.1) were detected and a large effect for parent-reported reductions in concerns about their child (CAMC Problem 1, ES 1.2, CI 0.4 to 2.0, CAMC Problem 2, ES 1.3, CI 0.5

to 2.1). No effects were detected for parenting behaviour and adult mental health. Descriptive scrutiny of follow-up findings showed that outcome scores across both groups were generally maintained or continued to improve.

Estimates of SD and upper CIs (u80% CI) for future power calculations of main clinical outcomes are: ECBI Problem: SD 6.7 (u80% CI: 7.6), ECBI Intensity: SD 33.6 (u80% CI: 37.1), CAMC Problem 1: SD 15.3 (u80% CI: 17.4) and KPSS: 3.0 (u80% CI: 3.3).

### Health economic findings

Details of the health economic analyses are provided in full in Day *et al*. At Time 2 CSRI data were available for 26 cases and 19 cases at Time 3. CSRI Time 2 data revealed that the services most used by caregivers included general practitioners (GPs), psychiatrists, other medics and social workers. Caregivers were often in receipt of medication. Children were frequently in contact with school nurses, dentists, opticians and GPs. EQ-5D-5L data were available for 36 caregivers at Time 1, 32 at Time 2 and 21 at Time 3. EQ-5D-Y data were available for 38, 31 and 20 children at each of the respective time points.

### CONCLUSION

This study assessed feasibility, potential challenges and decision-making for a definitive HFP-M trial.[28–30 39] Participant identification methods were successful across a wide range of mental health and CSS teams, over 90% of referred parents met dual child and parent screening criteria. Non-diagnostic eligibility criteria were acceptable and effective in recruiting an appropriate, multi-morbid sample.[60 61] Negative life events and disrupted family functioning, characteristic of the sample population, commonly delayed screening, data collection and intervention delivery.[7] Nevertheless, participant retention exceeded the a priori primary feasibility criterion of 65%. Participants declining intervention conditions were less likely to be retained at follow-up. Participant recruitment was slower than planned, mainly due to operational difficulties in one site.

**Table 4** Parent reported child, parenting and parent clinical outcomes (intention to treat analysis)

| Measure | Group | Baseline mean (SD) (n) | Post-intervention mean (SD) (n) | Follow-up mean (SD) (n) | Baseline/post-intervention estimated mean difference (CI) | P value | Effect size |
|---|---|---|---|---|---|---|---|
| CAMC | | | | | | | |
| Problem 1 | Intervention | 84.6 (16.0) (n=24) | 45.2 (27.2) (n=18) | 58.2 (29.5) (n=13) | 18.633 (−40.177 to 2.910) | 0.087 | 1.2 (0.4–2.0) |
| | Usual care | 85.9 (14.7) (n=22) | 63.1 (31.1) (n=14) | 53.3 (34.4) (n=8) | | | |
| Problem 2 | Intervention | 85.0 (19.1) (n=24) | 51.1 (31.9) (n=18) | 56.4 (35.5) (n=13) | 22.486 (−44.318 to −0.654) | 0.044 | 1.3 (0.5–2.1) |
| | Usual care | 82.9 (15.7) (n=22) | 72.5 (26.0) (n=14) | 66.5 (35.5) (n=8) | | | |
| Problem 3 | Intervention | 81.0 (17.0) (n=24) | 54.4 (33.8) (n=17) | 42.8 (35.4) (n=13) | 2.909 (−28.349 to 22.530) | 0.816 | 0.0 (−0.7–0.7) |
| | Usual care | 78.9 (18.5) (n=22) | 57.3 (34.1) (n=14) | 39.6 (24.3) (n=8) | | | |
| ECBI | | | | | | | |
| Problem score | Intervention | 22.1 (7.4) (n=24) | 17.9 (8.5) (n=15) | 15.3 (9.4) (n=13) | 1.559 (−4.24=55 to 7.374) | 0.585 | 0.1 (−0.7–0.88) |
| | Usual care | 22.43 (6.2) (n=21) | 18.0 (11.5) (n=14) | 18.8 (12.3) (n=8) | | | |
| Intensity score | Intervention | 168.2 (35.9) (n=24) | 142.4 (39.3) (n=24) | 131.7 (43.0) (n=18) | 12.866 (−4.24=55 to 7.374) | 0.233 | 0.4 (−0.3–1.1) |
| | Usual care | 169.5 (31.8) (n=23) | 155.7 (49.6) (n=14) | 148.9 (58.4) (n=8) | | | |
| CBCL-Int (t-score) | Intervention | 72.9 (9.7) (n=24) | 69.6 (10.4) (n=18) | 68.2 (8.3) (n=13) | 1.853 (−9.023 to 5.317) | 0.601 | 0.2 (−0.5–0.9) |
| | Usual care | 70.9 (11.9) (n=23) | 70.9 (9.3) (n=14) | 69.6 (8.6) (n=8) | | | |
| KPSS | Intervention | 10.1 (3.1) (n=24) | 12.9 (3.5) (n=18) | 14.9 (3.8) (n=13) | 1.177 (−1.260 to 3.615) | 0.331 | 0.4 (−0.3–1.1) |
| | Usual care | 10.9 (2.8) (n=23) | 12.4 (3.9) (n=14) | 13.6 (3.9) (n=7) | | | |
| PS | Intervention | 111.0 (19.6) (n=24) | 108.7 (16.5) (n=18) | 98.3 (27.2) (n=13) | 0.210 (−14.776 to −15.196) | 0.977 | 0.0 (−0.7–0.7) |
| | Usual care | 113.5 (24.8) (n=23) | 108.7 (28.9) (n=14) | 98.1 (26.9) (n=7) | | | |
| SCL-27 Global Severity Index | Intervention | 1.8 (1.1) (n=24 | 1.6 (0.8) (n=18) | 1.9 (0.8) (n=13) | −0.105 (−0.599 to 0.389) | 0.666 | −0.1 (−0.8–0.6) |
| | Usual care | 1.7 (0.8) (n=22) | 1.7 (0.9) (n=14) | 1.3 (1.0) (n=8) | | | |

CAMC, Concerns About My Child; CBCL-Int, Child Behaviour Checklist-Internalising Scale; ECBI, Eyberg Child Behaviour Inventory; KPSS, Kansas Parental Satisfaction Scale; PS, Arnold-O'Leary Parenting Scale; SCL-27, Symptom-Checklist-27.

Participants' multimorbid characteristics are commonly associated with poorer treatment engagement and outcomes.[62 63] Intervention uptake differed substantially between trial conditions. HFP-M was largely acceptable to participants but delivery was less efficient than planned, often due to parents' life circumstances. The augmented Usual care condition appeared to be less acceptable, and potentially affected participant retention. Lower Usual care retention and acceptability rates may reflect common dissatisfaction associated with control condition allocation.[64 65]

Clinical effects were detected across trial conditions, with potential advantage for child behaviour, parental child concerns and parenting satisfaction for HFP-M.

These are welcome given the population's complex parenting impairments and negative treatment expectancies.[2 63] The final sample size limited trial power and consequently affected interpretation of results. A definitive trial could potentially narrow child selection criteria to include only behaviour problems as parenting programmes have a stronger evidence base for this condition. Alternatively, HFP-M content may require strengthening specifically in relation to child internalising difficulties. Economic findings indicated potential cost advantages for HFP-M over usual care.[43] However, CSRI completion rates were not high and a simpler version may be required in a larger trial.

Trial conditions may have differed in duration, location and therapeutic intensity, potentially accounting for outcome and acceptability differences. Data collection relied on parent self-report, which is conventional given poor reliability of child-report across the age group and costs associated with independent ratings. Trial therapists provided self-report intervention fidelity data and supervision examined therapist HFP-M skills and implementation. Independent methods could strengthen validity of fidelity monitoring in a definitive trial, including observational and video methods.

A definitive trial is potentially feasible and should be based on the assumption of a medium effect size for the primary outcome of child behaviour. Site engagement, resource allocation and keyworker training in participant identification and recruitment will be crucial to enrolment of the larger sample required in a future trial. Embedded researchers assisting in caseload identification and direct parent recruitment, not possible in this feasibility study, may promote enrolment. Participant retention, particularly parents allocated to a usual care condition, will continue to be challenging for a full trial. The population's complex personality difficulties and, typically, heightened sensitivity to rejection, underline the importance of managing sensitively and effectively trial consent procedures, the emotional and practical consequences of random allocation and proactively maintaining communication and validating relationships with participants throughout trial duration, particularly for those allocated to usual care. Though not routinely available, the usual care condition could be augmented with an ongoing parent support group to potentially increase equipoise, face validity and uptake, which may also benefit trial retention.

HFP-M clinical and trial efficiency may be improved with more explicit, validating discussion with participants about the potential impact of life and personal circumstances on attendance, use of pre-emptive cancellation plans and inclusion of inter-session contact using digital technology.[31 63 66]

**Author affiliations**
[1]Department of Psychology, Institute of Psychiatry, Psychology and Neuroscience, King's College, London, UK
[2]Centre for Parent and Child Support, South London and Maudsley NHS Foundation Trust, London, UK
[3]The Centre for Psychiatry, Imperial College, London, UK
[4]McPin Foundation, London, UK
[5]King's Health Economics, PO24 David Goldberg Centre, Institute of Psychiatry, Psychology & Neuroscience, King's College London, London, UK
[6]Centre for Mental Health, Institute for Lifecourse Development, University of Greenwich, London, UK
[7]Institute of Mental Health, University of Nottingham, Nottingham, UK
[8]School of Psychology, University of Sussex, Brighton, UK
[9]Department of Population Health Sciences, Centre for Academic Mental Health, University of Bristol, Bristol, UK
[10]Department of Child and Adolescent Psychiatry, Institute of Psychiatry, Psychology and Neuroscience, King's College London, London, UK
[11]Department of Biostatistics, Institute of Psychiatry, Psychology and Neuroscience, King's College London, London, UK
[12]Faculty of Education, PEDAL Research Centre, University of Cambridge, Cambridge, UK
[13]Department of Mental Health, Middlesex University, London, UK

**Acknowledgements** We are indebted to the parents, practitioners, clinicians and service managers who worked with us on this trial, particularly Ruth Wilson. We would particularly like to thank Sarah Inkpen, Bethan Stevenson and Chelsea McCorry for their help with trial findings and Peter Fonagy and Philip Graham for overseeing the governance of the trial.

**Contributors** CD, Head, Centre for Parent and Child Support, South London, and Maudsley NHS Foundation Trust and Head, CAMHS Research Unit, King's College, London, was the chief investigator and responsible for the overall conception, design, data acquisition, analysis, interpretation of findings at each phase of this research. He is responsible for the overall content of this report. JB, Department of Psychology, King's College, London, was the senior research trial coordinator and was responsible for data acquisition, analysis, interpretation, drafting findings and revising this report. MJC, Professor in Mental Health Research, Centre for Psychiatry, Imperial College, London, was a co-applicant and contributed to the overall conception, design and interpretation of findings at each phase of this research. He provided an expert contribution about personality disorder. He contributed to critically revising this report. LF, McPin Foundation, was a service user researcher and representative in this research. She contributed to the design, data acquisition, analysis, interpretation and drafting of findings. She contributed to critically revising this report. LH, Deputy Head, Centre for Parent and Child Support, South London and Maudsley NHS Foundation Trust, was responsible for the development and revision of Helping Families Programme-Modified and supervised trial therapists. She contributed to critically revising this report. JB is a researcher at King's Health Economics, King's College, London, co-conducted the health economic analysis and drafted findings for this report. PM, Professor of Health Economics and Director of King's Health Economics, King's College, London, was a co-applicant and contributed to the overall conception, design and interpretation of findings at each phase of this research. He led the health economic component of the research and contributed to critically revising this report. MM, Professor at the Institute of Mental Health, University of Nottingham, was a co-applicant and contributed to the overall conception, design and interpretation of findings at each phase of this research. She provided an expert contribution about personality disorder. She was a member of the Manualisation Working Group, contributed to research therapist training and contributed to critically revising this report. DM, Senior Lecturer in Clinical Psychology, School of Psychology, University of Sussex, was a co-applicant and worked closely with the chief investigator on the overall conception and design of this research. He was senior research coordinator during the intervention development phase leading up to this trial and contributed to critically revising this report. PM, Professor of Psychiatry, Population Health Sciences, University of Bristol, was a co-applicant and contributed to the overall conception, design and interpretation of findings at each phase of this research. He provided an expert contribution about personality disorder. He contributed to research therapist raining and critically revised this report. LM, CAMHS Research Unit, King's College, London, was a researcher worker on this study. She was responsible for Phase 3 data acquisition, analysis and interpretation of findings. She completed the majority of qualitative data acquisition and led its analysis under the direction of the chief investigator. She contributed to critically revising this report. SS, Professor of Child Health and Behaviour, King's College, London, was a co-applicant and contributed to the overall conception, design and interpretation of findings at each phase of this research. He provided an expert contribution about parenting and parenting programmes. He contributed to critically revising this report. DS, Reader in Biostatistics, Biostatistics and Health Informatics, King's College, London, was a co-applicant and contributed to the overall conception, design and interpretation of findings at each phase of this research. He provided an expert contribution to the design, analysis and interpretation of quantitative components of the research. He contributed to critically revising this report. PR, LEGO Professor of Play in Education, Development and Learning, Faculty of Education, University of Cambridge, was a co-applicant and contributed to the overall conception, design and interpretation of findings at each phase of this research. He provided an expert contribution about child mental health and co-ordinated the research in CNWL. He contributed to critically revising this report. TW, Associate Professor in Mental Health, Mental Health Social Work and Interprofessional Learning, Middlesex University, London, was a co-applicant and contributed to the overall conception, design and interpretation of findings at each phase of this research. He provided an expert contribution about qualitative design, data acquisition and interpretation. He contributed to critically revising this report.

**Funding** This work was supported by National Institute of Health Research, Health Technology Assessment, Project Reference Number: 12/194/01.

**Competing interests** CD is the lead developer of two parenting programmes used in this report: Helping Families Programme and Empowering Parents Empowering Communities. MJC has previously received research grant funding on behalf of Imperial College London from the National Institute for Health Research. LH is a co-developer of the Helping Families Programme. MM was an author of the Psychoeducation plus Problems Solving (PEPS) intervention for adults with personality disorder. PEPS helped to inform the modified HFP. PM reports personal fees from a talk given at Fourth Bergen International Conference on Forensic Psychiatry 2016, Outside of the submitted work. PM led the development of the SAPAS, the personality disorder screen used in this study.

**Patient consent for publication** Not required.

**Ethics approval** Ethics approval was obtained from Health Research Authority South East Coast - Brighton and Sussex Research Ethics Committee (reference: 16/LO/0199).

**Provenance and peer review** Not commissioned; externally peer reviewed.

**Data availability statement** Data are available upon reasonable request. All manuals can be obtained from the corresponding author. All data requests should be submitted to the corresponding author for consideration. Access to anonymised data may be granted following review.

**ORCID iD**
Crispin Day http://orcid.org/0000-0002-7655-7839

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
