## [Reviewer comments · BMJ Open]

ARTICLE DETAILS

TITLE (PROVISIONAL)	Randomised Feasibility Trial of the Helping Families Programme-Modified: An Intensive Parenting Intervention for Parents Affected by Severe Personality Difficulties
AUTHORS	Day, Crispin; Briskman, Jackie; Crawford, Mike; Foote, Lisa; Harris, Lucy; Boadu, Janet; McCrone, Paul; McMurrin, Mary; Michelson, Daniel; Moran, Paul; Mosse, Liberty; Scott, Stephen; Stahl, Daniel; Ramchandani, Paul; Weaver, Tim

VERSION 1 – REVIEW

REVIEWER	Anne Sved Williams University of Adelaide, Department of Psychiatry
REVIEW RETURNED	15-Sep-2019

GENERAL COMMENTS	Parenting intervention reviewer comments Thanks for inviting me to review this paper. The authors are attempting to tackle an extremely difficult problem – intergenerational transfer of problems from personality-disordered mother (or father in one case) in offspring who have already begun to show significant effects of compromised parenting. Families with these problems are often non-attenders who distrust services and live with a great deal of compromise and chaos. The costs of helping both mother and offspring as the years go by are enormous so well worth tackling from compassionate and economic grounds. Thus the plan which is spelled out in an earlier paper by the authors is necessarily complex and well -structured but the results obtained rather disappointing especially in terms of enrolment. Their article is very direct about the problems they have run into during the course of this trial and they have proposed what appear to be realistic solutions to the identified issues. The article is certainly publishable in this journal and I think it is appropriate to highlight to a general audience that early intervention may potentially change life trajectories and help stop some children moving into adult mental health services as well as unemployment etc. The problems these families face are evident to all those working in health. Given the nature and scale of the problem and the involvement of health economists, I was disappointed to see no summary overview of the findings on economic benefits. I see that the authors have separated out the economic effects for a separate paper (a focus on economics is stated very clearly as one of their goals in their first paper where they outline their proposal) and that that paper has been accepted for publication. However as I cannot
---

	see that paper as it is in press, it would be relevant to know a little more in this paper so that a more general audience who may only read one paper on this topic can see the intervention's economic value more clearly. Secondly, nowhere do they mention that perhaps intervening with children aged 3-11 might already be a long way down the track for the children, Again, given the extent of the problem, the very few published attempts to intervene and the generalist nature of the presumed readership of BMJOpen if it is to be published in this journal, some discussion about age of child at intervention may be appropriate. Reading this paper makes me want to redouble my efforts in the perinatal and infant area to intervene before the infant/child problems are too entrenched in the child – and on that note it was disappointing that the authors didn't quote our paper which provides detail of one of the extremely rare attempts to tackle both maternal BPD functioning and parenting. (Sved Williams et al, A new therapeutic group to help women with borderline personality disorder and their infants, J Psychi Prac 2018) – our results are not dissimilar to those in this paper Next, there is little focus on the fact that the active condition offers home visiting which increased both the likelihood of compliance (which is excellent of course) but makes it less comparable to some other interventions. Perhaps this issue is addressed in the economic paper but might be worth a mention, Lastly a couple of very minor issues noted in the text follow:  1. P9 para 3 “The Helping Families Programme-Modified (HFP-M) was developed as a specialist, intensive parenting interventions to address this need and service gap. – Incorrect grammar as it is “a” and then “interventions” 2. P 11 last para – ref missing
--	---

REVIEWER	Brin Grenyer University of Wollongong
REVIEW RETURNED	22-Oct-2019

GENERAL COMMENTS	This addresses the important area of parenting with personality disorder. The authors report considerable challenges in implementing the trial. A number of key areas are suggested to be addressed to improve the manuscript.  1. There are important differences in the actual trial reported, compared to the published protocol, and these should be discussed and addressed. Significantly, the protocol suggests SCID DSM interviews will be conducted to establish personality disorder, but the trial used the short screening SAPAS. Most consider a score of 4 to be the correct cut-off on the SAPAS, yet the authors choose 3. Which version of the SAPAS was used? It is really difficult to understand who was being treated for what - the title suggests "severe" personality disorder but that is not supported by this cut off and no other clinical information justifies or explains further what were the range of mental health diagnoses of participants.
--

2. Why for the inclusion criteria did the child need to be experiencing significant emotional/behavioural difficulties? The concern here is that it is unclear if the target for intervention was the parent or the child. The stated objective is parenting - thus presumably the goal would be both to improve parenting to also protect against future adversity in child emotional/behavioural functioning, rather than needing this upfront. The additional concern is one of causality - was the primary problem the child causing additional stress in vulnerable parents (thus suggesting the right target was not the parent but rather the child)?

3. The choice of intervention (skills based parenting) suggests this intervention was about upskilling the parent rather than increasing their emotional connection as is more typical of attachment-based interventions. The manualised approach selected needs further discussion and description. It is unclear how it is designed to specifically address personality disorder and parenting difficulties. For example, most consider attachment enhancing interventions are important, and these are cited by the authors, but it is unclear if these are a feature of the intervention. How the approach is similar or different to other published approaches should be discussed. Why were these three targets chosen and not others?

4. Why was an exclusion criterion (in the protocol) attendance by the parent in actual treatment (group or individual) for personality disorder? Is the thinking that treating the personality disorder would make redundant the intervention (if so, why not just treat the disorder), OR was the thinking that it might make it hard to determine if the intervention was effective (but the intervention is not targeted at personality disorder). There are ethical issues here - should parenting interventions only be for those untreated, OR is there thinking that treatments should be 'spread out' fairly in the population so if you have one you should not have the other? Other approaches suggest that parenting interventions should be "added on" to existing therapy for the actual disorder - why is that not done here? Personality disorder is a treatable condition - why would this intervention aimed at parenting not also seek to actually treat the disorder itself?

5. The study claims to be the 'first feasibility trial' in this area, which ignores the published work already done which should be considered e.g. DOI 10.1080/18387357.2018.1464887, 10.1371/journal.pone.0223038

6. Why were treatments conducted in the home? Most consider that an important factor leading to change is the activation of patient agency e.g. doi: 10.1016/S0140-6736(14)61394-5. This can include active attendance at health facilities. Clinician home visiting appears to be 'wrong touch' in terms of facilitating active rehabilitation and recovery principles despite being important in early engagement.

7. Further discussion of fidelity of implementation needs to occur. The protocol discussed ways that actual feasibility might be assessed, e.g. by the researcher visiting / observing interventions. It is difficult as it stands to know what was actually delivered in either intervention. The gold standard would be to videotape actual therapies and observers to rate videotapes on adherence and competence.

	8. It appears that of the 19 TAU only 6 provided follow-up data, meaning consideration should be given to removing all statistical analyses. 9. The manuscript has many errors and typos Examples: p.3 line 29 "specialist parenting." ...? p. 6 therapist "raining" Reference 61 is underlined
--	---

REVIEWER	Dr Linda Williams University of Edinburgh, UK
REVIEW RETURNED	11-Nov-2019

GENERAL COMMENTS	In terms of statistical analysis, I am happy with what has been written. However, given the modest target 70 participants recruited was not met, the poor acceptance of the standard care following randomisation to it, and the poor attendance even in the new treatment, I feel the authors are rather over-optimistic in their conclusions. I would prefer to see more caution expressed.
---

VERSION 1 – AUTHOR RESPONSE

Reviewer(s)' Comments to Author:

Reviewer: 1

Reviewer Name: Anne Sved Williams

Institution and Country: University of Adelaide, South Australia; Women's and Children's Health Network, South Australia Please state any competing interests or state 'None declared': none declared

Please leave your comments for the authors below Parenting intervention reviewer comments

Thanks for inviting me to review this paper. The authors are attempting to tackle an extremely difficult problem – intergenerational transfer of problems from personality-disordered mother (or father in one case) in offspring who have already begun to show significant effects of compromised parenting. Families with these problems are often non-attenders who distrust services and live with a great deal of compromise and chaos. The costs of helping both mother and offspring as the years go by are enormous so well worth tackling from compassionate and economic grounds. Thus the plan which is spelled out in an earlier paper by the authors is necessarily complex and well -structured but the results obtained rather disappointing especially in terms of enrolment. Their article is very direct about the problems they have run into during the course of this trial and they have proposed what appear to be realistic solutions to the identified issues. The article is certainly publishable in this journal and I think it is appropriate to highlight to a general audience that early intervention may potentially change life trajectories and help stop some children moving into adult mental health services as well as unemployment etc. The problems these families face are evident to all those working in health.

We appreciate the reviewer's comments and acknowledgement of the practical challenges of conducting intervention research with this target population. There were significant differences in participant enrolment across the two trial sites, with one site recruiting significantly more participants. We refer to this in the conclusion, see Manuscript Page 20, Paragraph 1. No amendments have been made to the manuscript text.

Given the nature and scale of the problem and the involvement of health economists, I was disappointed to see no summary overview of the findings on economic benefits. I see that the authors have separated out the economic effects for a separate paper (a focus on economics is stated very clearly as one of their goals in their first paper where they outline their proposal) and that that paper has been accepted for publication. However as I cannot see that paper as it is in press, it would be relevant to know a little more in this paper so that a more general audience who may only read one paper on this topic can see the intervention's economic value more clearly.

We thank the reviewer for her interest in the trial's health economic findings. In original manuscript, we found it difficult to include an adequate summary of the health economic findings within the word limit. In response to the reviewer, we have elected to include a summary of findings. This is necessarily a summary with the main findings available in the trial's full report, which has been accepted for publication by the UK National Institute for Health Research.

The manuscript has been amended in the following ways:

Measures:

'Health economic:

- **Client Service Receipt Inventory (CSRI)**, [56] a schedule adapted to measure the use of services by caregivers and children.
- **EQ-5D-5L and EQ-5D-Y**, [48,49] a generic measure of health-related quality of life used to generate quality-adjusted life years. EQ-5D-Y is adapted for younger respondents. '

The following references have been added:

Beecham J, Knapp M. Costing psychiatric interventions. In Thornicroft G (ed) Measuring Mental Health Needs. London, Gaskell.

Wille, N., Badia, X., Bonsel, G., Burstro, K., Cavrini, G., Develin, N., et al. (2010). Development of the EQ-5D-Y: A child-friendly version of the EQ-5D. Qual Life Res, 19 (6), 875-86.

'Health economic findings

Details of the health economic analyses are provided in full in Day et al. At Time 2, CSRI data were available for 26 cases but only 19 at time 3. The data recorded with the CSRI at Time 2 revealed that the services most used by caregivers included GPs, psychiatrists, other doctors, and social workers. Caregivers were also often in receipt of medication. Children were frequently in contact with school nurses, dentists, opticians, and GPs. EQ-5D-5L data were available for 36 caregivers at Time 1, 32 at Time 2 and 21 at Time 3. EQ-5D-Y data were available for 38, 31 and 20 children at each of the respective time points.'

Conclusion, Manuscript Page 20.

'Economic findings indicated potential cost advantages for HFP-M over usual care.[44] However, CSRI completion rates were not high and a simpler version may be required in a larger trial.'

3. **Secondly, nowhere do they mention that perhaps intervening with children aged 3-11 might already be a long way down the track for the children, Again, given the extent of the problem, the very few published attempts to intervene and the generalist nature of the presumed readership of BMJOpen if it is to be published in this journal, some discussion about age of child at intervention may be appropriate. Reading this paper makes me want to redouble my efforts in the perinatal and infant area to intervene before the infant/child problems are too entrenched in the child – and on that note it was disappointing that the authors didn't quote our paper which provides detail of one of the extremely rare attempts to tackle both maternal BPD functioning and parenting.(Sved Williams et al, A new therapeutic group to help women with borderline personality disorder and their infants, J Psychi Prac 2018) – our results are not dissimilar to those in this paper**

We agree with the reviewer that there is evidence of parenting and child development difficulties in the target population from infancy. Preventive interventions from pregnancy and infancy are warranted and are currently being evaluated (see Sved Williams et al, 2018). The trial was awarded funding by the UK National Institute of Health Research (NIHR) Health Technology Assessment in response to specific commissioned call for a research programme to develop and evaluate psychoeducational support for parents with personality disorders whose children had severe emotional and behavioural problems and who were attending, or being considered for referral to, child and adolescent mental. Our intention with this trial was therefore to examine the feasibility of clinical intervention with the target population where mental health difficulties in the parent and child were established. health services. In doing so, we wanted to make use of and extend the established evidence base for parenting interventions for children aged 3-11years (NICE, 2013).

National Institute for Health and Clinical Excellence (2013). *Antisocial behaviour and conduct disorders in children and young people: Recognition, intervention and management*. London: NICE.

We have amended the text on Manuscript Page 8, Paragraph 4 in response to the reviewer's comments as follows:

'Concerted preventative and early intervention during pregnancy, infancy and childhood is warranted. [16]'

The following reference has been added

[16] Sved Williams, A., Yelland, C., Hollamby, S., Wigley, M. Aylward, P. (2018) A new therapeutic group to help women with borderline personality disorder and their infants, *J Psychi Prac*, 24 (5): 331-340

Next, there is little focus on the fact that the active condition offers home visiting which increased both the likelihood of compliance (which is excellent of course) but makes it less comparable to some other interventions. Perhaps this issue is addressed in the economic paper but might be worth a mention,

Feasibility results indicated that the intervention condition had higher levels of acceptability. This may have been due to the intervention itself as well as confounding factors to which we refer on Manuscript Page 20: Conclusion.

Both the HFT-M and the additional single parenting session that augmented Usual Care were delivered in the home. Unfortunately, we did not collect data on the location of concurrent adult

mental health, social service and other care received by trial participants. We accept that the intervention participants may have received a greater amount of home-based intervention.

We have amended the text on Manuscript Page 20, Paragraph 4 to include reference to intervention location as follows:

'Trial conditions may have differed in duration, location and therapeutic intensity, which may account for potential outcome and acceptability differences.'

Lastly a couple of very minor issues noted in the text follow:

- 1. P9 para 3 "The Helping Families Programme-Modified (HFP-M) was developed as a specialist, intensive parenting interventions to address this need and service gap. – Incorrect grammar as it is "a" and then "interventions"**
- 2. P 11 last para – ref missing**

We have corrected all typographical errors. Reference not required

Reviewer: 2

Reviewer Name: Brin Grenyer

Institution and Country: Illawarra Health and Medical Research Institute, School of Psychology, University of Wollongong Australia.

Please state any competing interests or state 'None declared': None declared

Please leave your comments for the authors below This addresses the important area of parenting with personality disorder. The authors report considerable challenges in implementing the trial. A number of key areas are suggested to be addressed to improve the manuscript.

1. There are important differences in the actual trial reported, compared to the published protocol, and these should be discussed and addressed. Significantly, the protocol suggests SCID DSM interviews will be conducted to establish personality disorder, but the trial used the short screening SAPAS. Most consider a score of 4 to be the correct cut-off on the SAPAS, yet the authors choose 3. Which version of the SAPAS was used?

The eligibility criteria in the published trial protocol (Day et al. 2017) and this paper are the same. In our original funding application, we proposed to use SCID-II and DAWBA structured interviews as screening instruments because our funders were interested in the study being open to participants with and without pre-existing personality disorder diagnosis who fulfilled trial eligibility criteria. These instruments were used in the pre-feasibility trial case series reported in Wilson et al (2018).

Wilson, R., Weaver, T., Michelson, D. & Day, C. (2018) Experiences of parenting and clinical intervention for mothers affected by personality disorder: a pilot qualitative study combining parent and clinician perspectives. *BMC Psychiatry*, 18,152. doi.org/10.1186/s12888-018-1733-8.

Consultation with service users and clinicians in preparation for the trial, results from our pre-feasibility case series and consultation with research ethics committee indicated that initial plans for trial recruitment based on research diagnosis of personality disorder, using the Structured Clinical Interview for DSM-IV Axis II Disorders (SCID II), was unlikely to be viable (Wilson et al., 2018). Research ethics raised concerns about the implications of participants with no previous diagnosis of personality disorder acquiring a research diagnosis through the trial screening procedures. Participant feedback from the pre-trial feasibility case series indicated that the SCID II/DAWBA screening procedures were lengthy, burdensome and likely to impede recruitment. Researchers during field testing also reported that the planned screening procedures were considerably more lengthy than initially estimated, with significant time and cost implications for the trial. On this basis we revised the eligibility criteria to those in our trial protocol and this paper. This revision to the original research plan was approved by the Trial Steering Committee and NIHR funders prior to production of the trial protocol.

We have amended the text on Manuscript Page 10, Paragraph 3 as follows:

'Pre-feasibility trial case series findings, consultation with service user, clinicians and research ethics indicated that initial plans for trial recruitment based on personality disorder research diagnosis was unlikely to be viable for practical, participant acceptability and ethical reasons.[21] '

The trial recruited parents experiencing enduring and pervasive interpersonal and personality difficulties, without requiring a formal research diagnosis, assessed by a score of 3 or more on the self-administered Standardised Assessment of Personality - Abbreviated Scale (SAPAS). Professor Paul Moran, developer of the SAPAS, was a co-applicant for the trial and is a co-author of this paper.

The reviewer assumption about the 'correct' cut-point of 4 for the SAPAS is not accurate. The cut-point that optimises sensitivity and specificity of the SAPAS depends on the nature of the sample population. In non-clinical population samples, a cut-point of 4 optimises sensitivity and specificity (see Fok et al, 2015). Our previous work (Moran et al, 2003) established that 3 is the optimal cut-point for a clinical population such as the one used in this study.

We have made the following amendment to the manuscript on Manuscript Page 11, Paragraph 3: Eligibility criteria:

'(iii) experiencing severe personality difficulties, assessed by self-administered Standardised Assessment of Personality-Abbreviated Scale (SAPAS) score of ≥ 3 , the optimal cut-point for the intended sample population, [45]'

Fok ML, Seegobin S, Frissa S, Hatch SL, Hotopf M, Hayes RD, Moran P. Validation of the standardised assessment of personality-abbreviated scale in a general population sample. *Personality and mental health*. 2015 Nov;9(4):250-7.

Moran P, Leese M, Lee T, Walters P, Thornicroft G, Mann A. (2003) The Standardised Assessment of Personality - abbreviated scale (SAPAS): preliminary validation of a brief screen for personality disorder. *British Journal of Psychiatry*, 183(3), 228-232.

2. It is really difficult to understand who was being treated for what - the title suggests "severe" personality disorder but that is not supported by this cut off and no other clinical information justifies or explains further what were the range of mental health diagnoses of participants.

In clarification, we have used the term 'severe personality difficulties' throughout the paper, which we define and describe on Manuscript Page 8, Paragraph 2. We have not used the term 'severe

personality disorder' at any point . We used the well-validated SAPAS for screening personality difficulties and SDQ for screening child mental health difficulties not clinical or research diagnostic status.

The participants and their index child recruited into the trial met the caseness criterion on each of these instruments. We have not made any amendments to the manuscript in response to this comment.

3. Why for the inclusion criteria did the child need to be experiencing significant emotional/behavioural difficulties? The concern here is that it is unclear if the target for intervention was the parent or the child. The stated objective is parenting - thus presumably the goal would be both to improve parenting to also protect against future adversity in child emotional/behavioural functioning, rather than needing this upfront. The additional concern is one of causality - was the primary problem the child causing additional stress in vulnerable parents (thus suggesting the right target was not the parent but rather the child)?

Harsh, emotionally uninvolved, ambivalent and inconsistent parenting behaviours are associated with poor child emotional and behavioural outcomes. Structured skills-based parenting interventions are generally effective at improving parent and child outcomes. However, families with established intergenerational mental health difficulties are less likely to benefit from non-specialised parenting programmes.

The intention of this study was to assess the feasibility of Helping Families Programme-Modified, a specialised parenting intervention that targets families with co-existing parent personality difficulties and child mental health difficulties. The primary clinical outcome was child behaviour, with a range of secondary child, parenting and parent outcomes.

This rationale is set out on Manuscript Page 9, Paragraphs 3-4. The intervention was not intended as a preventive intervention for children 'at risk' of developing difficulties but for families with established child and parent difficulties as measured by the SAPAS and SDQ, as required by the NIHR Health Technology Assessment commissioned funding call.

We have inserted the following sentence to make clear the purpose of this trial on Manuscript Page 10, Paragraph 4:

'The trial aimed to assess research and clinical feasibility of HFP-M for a target population with co-existing parent personality difficulties and child mental health difficulties with findings being used to inform the design of a full-scale trial. '

We have amended the first line of the Abstract as follows:

'Specialist parenting intervention could improve coexistent parenting and child mental health difficulties of parents affected by severe personality difficulties.'

4. The choice of intervention (skills based parenting) suggests this intervention was about upskilling the parent rather than increasing their emotional connection as is more typical of attachment-based interventions. The manualised approach selected needs further discussion and description. It is unclear how it is designed to specifically address personality disorder and parenting difficulties. For example, most consider attachment enhancing interventions are important, and these are cited by the authors, but it is unclear if these are a feature of the

intervention. How the approach is similar or different to other published approaches should be discussed. Why were these three targets chosen and not others?

The development of the HFP-M intervention followed recommended MRC, CONSORT and TIDiER frameworks and guidelines, see Manuscript Page 9, Paragraph 3. The aims, content, methods and format of the HFP-M were informed by evidence derived from previous research (Day et al., 2011), structured consultation with service users and clinicians from child mental health and personality disorder services, best practice and research evidence recommendations and subsequent field testing in a case series prior to the feasibility trial (Wilson et al., 2018).

The intervention combines a range of cognitive, behavioural, affective and relational skills and methods rather than being restricted to 'skills based parenting' as suggested by the reviewer.

We have amended Manuscript Page 9, Paragraph 3 and 4 as follows to provide a more detailed but succinct description of HFP-M:

'Consistent with other promising programmes aiming to improve parenting and child outcomes in high risk groups, HFP-M is based on a transtheoretical model of parenting drawing on attachment, social learning and cognitive-affective theories and methods.[34,35] HFP-M does not target personality difficulties *per se* but aims to improve the ways that these characteristics affect parenting behaviour, emotional regulation, parent-child relationships and lead to adverse child outcomes.'

'HFP-M has three structured components [36]: (i) Core Therapeutic Process: including partnership and goal-based methods to promote collaborative relational engagement, shared formulation, empathic parent validation and crisis management;[36] (ii) Parent Groundwork: including emotion-focussed, cognitive, behavioural and interpersonal strategies to manage parental emotional dysregulation and hostility while relating to their children and undertaking parenting tasks, and (iii) Parenting Strategies: including consistent use of positive parenting skills, such as, praise, consequences and limit setting, and relational and affective parenting methods such as emotionally responsive, warm care-giving and reflective function.'

.

The following references have been added, with all references renumbered:

[34] Harnett, P. & Dawe, S. (2008). Reducing Child Abuse Potential in Families Identified by Social Services: Implications for Assessment and Treatment. *Brief Treatment and Crisis Intervention*, 8(3), 226–235.

[35] Gray, A. Townsend, M. Bourke, M. & Grenyer, B. (2018) Effectiveness of a brief parenting intervention for people with borderline personality disorder: a 12-month follow-up study of clinician implementation in practice, *Advances in Mental Health*, 17:1, 33-43, DOI: [10.1080/18387357.2018.1464887](https://doi.org/10.1080/18387357.2018.1464887)

5. Why was an exclusion criterion (in the protocol) attendance by the parent in actual treatment (group or individual) for personality disorder? Is the thinking that treating the personality disorder would make redundant the intervention (if so, why not just treat the disorder), OR was the thinking that it might make it hard to determine if the intervention was effective (but the intervention is not targeted at personality disorder).

Our original funding application included the criterion to exclude parents concurrently engaged in individual or group psychotherapy directly related to personality disorder. This criterion was removed following systematic consultation with clinicians and service users prior to the development of the trial

protocol. This exclusion criterion is therefore not included in the trial's published protocol or in this paper. This revision to the original application was approved by the Trial Steering Committee and NIHR funders prior to production of the trial protocol.

6 There are ethical issues here - should parenting interventions only be for those untreated, OR is there thinking that treatments should be 'spread out' fairly in the population so if you have one you should not have the other? Other approaches suggest that parenting interventions should be "added on" to existing therapy for the actual disorder - why is that not done here? Personality disorder is a treatable condition - why would this intervention aimed at parenting not also seek to actually treat the disorder itself?

The ethical concerns raised by the reviewer may be justified if the intervention evaluated had been restricted to untreated parents or was intended to be 'spread out fairly in the population'. These strictures were never considered by the research team and are not proposed in any of our work. The strictures did not apply to this trial and intervention. HFP-M was provided as an 'add-on' to existing treatment as usual.

Manuscript Page 14, Paragraph 3: Concomitant interventions has been amended to improve clarity as follows:

'Both HFP-M and the single Usual care parent support session were provided in addition to existing medical, psychosocial and educational support and treatment services used by participating parents and their families. A joint-working protocol specified procedures for care co-ordination and information sharing between trial therapists and routine services.'

7. The study claims to be the 'first feasibility trial' in this area, which ignores the published work already done which should be considered e.g. DOI 10.1080/18387357.2018.1464887, 10.1371/journal.pone.0223038

The reviewer is right to draw attention to our poorly drafted bullet point. It was not our intention to minimise the valuable work of the reviewer and other similar groups internationally. We have redrafted the bullet point to more accurately reflect our meaning as follows:

'This randomised trial assessed the feasibility of a specialist parenting intervention for coexistent mental health problems of parents affected by severe personality difficulties and their children.'

8. Why were treatments conducted in the home? Most consider that an important factor leading to change is the activation of patient agency e.g. doi: 10.1016/S0140-6736(14)61394-5. This can include active attendance at health facilities. Clinician home visiting appears to be 'wrong touch' in terms of facilitating active rehabilitation and recovery principles despite being important in early engagement.

We agree with the reviewer about the importance of activating patient agency in treatment approaches for this population. The relational, goal-orientated and progress monitoring components of HFP-M Core Therapeutic Process (see Manuscript Page 9, Paragraph 4) were specifically designed to activate parent agency.

The home and community-based delivery was intended to lower barriers to care, increase engagement and promote patient activation and agency. The three inter-related components of HFP-M (see Manuscript Page 9, Paragraph 4) are intended to mobilise change in parenting functioning and skills, active implementation of manualised strategies in the home and throughout family life, and proactive management of concurrent life crises. WAI-SR data reported in the paper provided evidence that supported the home and community-based approach as do the qualitative findings

reported in the NIHR report (Day et al, in press). We did not find qualitative or quantitative evidence from participants or their keyworkers in this trial or previous studies to suggest that the intervention reduced agency (Day et al., 2011).

Day, C., Kowalenko, S., Ellis, M., Dawe, S., Harnett, P. & Scott, S. (2011). The Helping Families Programme: A new parenting intervention for children with severe and persistent conduct problems. *Child Adol Ment H-UK*, 16, 167-171.

Day C, Briskman J, Crawford MJ, Foote L, Harris L, Boadu J, et al. Helping Families Programme-Modified: Development of a Specialised Parenting Intervention for Parents Affected by Severe Personality Difficulties and Randomised Feasibility Trial. *Health Technol Assess. Accepted for publication.*

Wilson, R., Weaver, T., Michelson, D. & Day, C. (2018) Experiences of parenting and clinical intervention for mothers affected by personality disorder: a pilot qualitative study combining parent and clinician perspectives. *BMC Psychiatry*, 18,152. doi.org/10.1186/s12888-018-1733-8.

9. Further discussion of fidelity of implementation needs to occur. The protocol discussed ways that actual feasibility might be assessed, e.g. by the researcher visiting / observing interventions. It is difficult as it stands to know what was actually delivered in either intervention. The gold standard would be to videotape actual therapies and observers to rate videotapes on adherence and competence.

We agreed that intervention fidelity is important to ensure consistent intervention delivery. We have added more detail about fidelity in the Method, see Manuscript Page 12, Paragraph 1 as follows:

‘Six trial therapists received eight, three-hour, training sessions provided by HFP-M programme developers and clinical experts. Trial therapists completed structured checklists and received fortnightly supervision from experienced HFP-M clinicians to support clinical implementation and fidelity.’

We have added the following sentence on Manuscript Page 21, Paragraph 1: Conclusion to reflect this:

‘Trial therapists provided self-report intervention fidelity data and supervision examined therapist HFP-M skills and implementation. Independent methods could strengthen validity of fidelity monitoring in a definitive trial, including observational and video methods.’

10. It appears that of the 19 TAU only 6 provided follow-up data, meaning consideration should be given to removing all statistical analyses.

24 parents were randomised to the Usual Care condition, of whom 14 provided post-intervention follow-up data and 8 provided 4-month follow-up data. Allocation to the Usual Care condition was a major reason for participant attrition. Inclusion of the 4-month follow-up data reinforces this learning from the feasibility trial and the underlines the implications for the definitive trial. We have not amended the manuscript text.

9. The manuscript has many errors and typos

Examples:

p.3 line 29 "specialist parenting." ...?

p. 6 therapist "raining"

Reference 61 is underlined

All typographical errors have been corrected

Reviewer: 3

Reviewer Name: Dr Linda Williams

Institution and Country: University of Edinburgh, UK Please state any competing interests or state 'None declared': None declared

Please leave your comments for the authors below In terms of statistical analysis, I am happy with what has been written. However, given the modest target 70 participants recruited was not met, the poor acceptance of the standard care following randomisation to it, and the poor attendance even in the new treatment, I feel the authors are rather over-optimistic in their conclusions. I would prefer to see more caution expressed.

This study was intended to examine the feasibility of intervention delivery and trial procedures with a view to a future definitive trial of HFP-M. The study did find evidence to support the primary feasibility criterion (i.e., a post-intervention retention rate above 65%). We agree with Dr Williams's caution about the feasibility challenges of recruitment, which were caused by recruitment delays at one of the two trial sites. This is referred to in the Conclusion, Manuscript Page 20, Paragraph 1. The study also helpfully identified questions about the acceptability of the augmented Usual Care condition and implications for participant retention for this condition in particular. These feasibility trial findings have assisted the research team to plan ways to address these issues in a definitive trial. We have outlined these recommendations in the Conclusion.

We have amended the text at two points to reflect Dr Williams's request for additional caution as follows:

Conclusion Manuscript Page 21, Paragraph 2:

'A future definitive trial is potentially feasible and should be based on the assumption of a medium effect size for the primary outcome of child behaviour.'

Article summary, bullet point 2:

'Findings provide useful evidence to support further evaluation of this specialist parenting intervention, including planning a definitive trial, with modifications required to improve intervention efficiency, augmented usual care condition acceptability, and participant enrolment and retention.'

In making these amendments, we have edited the manuscript text to incorporate the additional required text. The total word count for the manuscript is now 4227.

VERSION 2 – REVIEW

REVIEWER	Anne Sved Williams University of Adelaide, Australia
REVIEW RETURNED	29-Dec-2019

GENERAL COMMENTS	This is a very well written paper outlining an extremely well thought through intervention in a very troubled population whose needs are well worth addressing. The complexities of this population are well known and the authors have done well to provide a structured approach to therapy and outline the issues and outcomes so clearly. I understand that the statistics have already been reviewed which is why i have ticked the No box for statistical review.
---

REVIEWER	Brin Grenyer Illawarra Health and Medical Research Institute, School of Psychology, University of Wollongong Australia
REVIEW RETURNED	17-Jan-2020

GENERAL COMMENTS	Authors have improved the manuscript in their response to reviewer's comments
---

REVIEWER	Dr Linda Williams University of Edinburgh, UK
REVIEW RETURNED	19-Dec-2019

GENERAL COMMENTS	I am still somewhat concerned about the ability of a larger scale trial to retain sufficient patients on Usual care in order to be able to test for any differences, but that is more of an issue for the future and not this paper
---